# Elucidating the mechanistic association of xylene inducing non-small cell lung cancer through network toxicology and molecular docking analysis

Hongquan Chen[1,2☯], Xiangpeng Chen[3☯], Weibin Lin[1☯], Qing Chen[1], Renxi Lin[1], Mingfang Zhang[1]*, Yuanlin Qi[1]*

1 School of Basic Medical Sciences, Fujian Medical University, Fuzhou, Fujian, China, 2 National Key Laboratory of Intelligent Tracking and Forecasting for Infectious Diseases (NITFID), National Institute for Communicable Disease Control and Prevention, Chinese Center for Disease Control and Prevention, Beijing, China, 3 Graduate Training Base of General Hospital of Northern Theater Command of Dalian Medical University, Shenyang, China

☯ These authors contributed equally to this work.
* mfzhang@fjmu.edu.cn (MZ); ylqi@fjmu.edu.cn (YQ)

## Abstract

Xylene is a common industrial solvent that includes three isomers: o-xylene, m-xylene, and p-xylene. Long-term exposure to low doses of xylene in the environment has been linked to a higher risk of lung cancer. However, the molecular mechanisms behind this link are still not fully understood. In this study, we used a combination of network toxicology and molecular docking to investigate how xylene may contribute to the development of non-small cell lung cancer (NSCLC). We first identified 115 potential target genes related to xylene exposure by searching several public databases, including CHEMBL, STITCH, GeneCards, and OMIM. Further screening using the STRING platform and Cytoscape analysis highlighted five core targets: IL1A, H3C13, ITGAM, CCR5, and COMT. We utilized scRNA-seq data to analyze the expression patterns of core targets across distinct cell subpopulations, the majority of core targets were expressed in immune cells. We then performed GO and KEGG pathway enrichment analysis. These results showed that the five target genes are mainly involved in cancer-related pathways, such as ECM-receptor inter-action, focal adhesion, chemical carcinogenesis, and the PI3K-Akt signaling pathway. Molecular docking results confirmed that xylene isomers have strong binding affinities with the proteins encoded by these genes. This suggests that xylene may disrupt important cellular signals and promote tumor growth. In conclusion, our study provides new insight into how xylene might cause NSCLC at the molecular level. It also shows the usefulness of network toxicology in evaluating health risks from environmental chemicals. These findings may help guide future efforts in prevention and treatment strategies.

**Data availability statement:** Publicly available datasets were analyzed in this study and are noted in the paper.

**Funding:** The research was supported by the grants from the Natural Science Foundation of Fujian Province (2023J01300, 2025J01686), the Startup Fund for scientific research of Fujian Medical University (2021QH1002, 2022QH1005), and the Provincial Department of Education-Science and Technology (JAT210107). The funders had roles in study design, and decision to publish.

**Competing interests:** The authors have no competing interests to declare that are relevant to the content of this article.

## Introduction

Benzene, toluene, ethylbenzene, and xylene (BTX) are monocyclic aromatic volatile organic compounds (VOCs) that are widely produced globally [1]. Their major sources are petroleum refining and emissions from the petrochemical industry [2]. Because of their potential harm to human health, BTX compounds have been classified as hazardous air pollutants [3]. Among these, xylene holds a significant share in the global market and is extensively used as an organic solvent in products such as paints, dyes, thinners, and adhesives [4]. It is also a commonly used hazardous chemical in pathology laboratories worldwide [5]. Xylene exists in three structural isomers: ortho-xylene (o-xylene), meta-xylene (m-xylene), and para-xylene (p-xylene), which differ based on the position of the two methyl groups on the benzene ring at the 1,2, 1,3, and 1,4 positions, respectively [6]. Commercial xylene is typically a mixture of these isomers [7]. Because xylene is a volatile liquid, inhalation is the main route of exposure for humans. Short-term exposure can lead to irritation of the eyes, nose, and throat, as well as dysfunction in the nervous, digestive, and reproductive systems [6]. Long-term exposure may cause chronic damage to the respiratory system, central nervous system, cardiovascular system, and kidneys [6,7]. The lungs are especially vulnerable to xylene because they are the first target organ upon inhalation. Epidemiological studies have suggested a correlation between high xylene exposure and increased lung cancer incidence [8]. However, clear evidence for the underlying molecular mechanisms is still lacking.

Lung cancer remains the leading cause of cancer-related mortality worldwide, with more than 85% of cases classified as non-small cell lung cancer (NSCLC) [9]. While tobacco smoking is still considered the primary risk factor [10], the role of non-smoking-related factors in lung cancer development has become increasingly prominent in recent years. Epidemiological studies have predicted that lung cancer in non-smokers will rank as the fifth leading cause of cancer death globally [11]. The pathogenesis of lung cancer in non-smokers is closely linked to environmental pollutants (such as PM2.5 and polycyclic aromatic hydrocarbons), occupational exposure to toxic substances (such as BTX compounds and asbestos), and genetic susceptibility (such as EGFR mutations) [11,12]. Xylene is a common environmental and occupational pollutant, may contribute to lung cancer development due to its high volatility and tendency to accumulate in the body. Recent studies have confirmed that long-term exposure to low-level ambient BTX is significantly associated with an increased risk of lung, thyroid, prostate, and other common cancers [13]. The potential carcinogenicity of BTX involves multiple mechanisms, including oxidative stress-induced genetic toxicity, epigenetic dysregulation affecting DNA repair and gene expression, and disruption of the p53 pathway [13–15]. Therefore, investigating the molecular association between xylene exposure and NSCLC is of great importance for understanding its health risks and uncovering the mechanisms of environmentally induced carcinogenesis.

Network toxicology combines bioinformatics, big data, and multi-omics (genomics, proteomics, metabolomics) to map interactions between toxins, targets, and diseases, simplifying complex toxicity mechanisms into visual networks [16]. Molecular

docking, a computational drug design tool, predicts toxin-target interactions by analyzing binding affinity—lower binding energy indicates stronger, more stable binding [17]. Together, these methods systematically reveal toxicity mechanisms and key bioactive compounds. This study employed a network toxicology approach combined with molecular docking analysis to systematically investigate the potential toxic effects and underlying molecular mechanisms of xylene in the progression of NSCLC. By constructing a multi-layered regulatory network, the study aimed to identify possible pathogenic pathways of xylene from multiple perspectives and to elucidate its molecular basis of action. These findings provide a theoretical foundation for further understanding the toxicological characteristics of xylene and offer scientific support for the toxicity assessment of environmental pollutants and the prevention of related diseases.

## Methods

### Network toxicology analysis of xylene

The three isomers of xylene, o-xylene, m-xylene, and p-xylene, were first retrieved from the PubChem database (https://pubchem.ncbi.nlm.nih.gov/), and their corresponding SMILES structures were obtained. These SMILES sequences were then individually submitted to the ADMETlab 2.0 platform (https://admetmesh.scbdd.com/service/evaluation/cal) and the ProTox-II database (https://tox.charite.de/protox3/) to predict the toxicity profiles of xylene. All database searches for this study were conducted up to April 2025.

### Collection of xylene targets

Based on the SMILES structures of xylene retrieved from the PubChem database, potential target genes were predicted using the ChEMBL databas (https://www.ebi.ac.uk/chembl/) and the STITCH database (http://stitch.embl.de/). The ChEMBL identifiers and target information obtained from STITCH were integrated and duplicates were removed. Target names were then standardized using information from the UniProt database, resulting in the construction of a xylene target gene set.

### Collection of targets associated with NSCLC

By searching for the keyword "non-small cell lung cancer" in the GeneCards database (Version 5.19) and the OMIM database, target information related to NSCLC was collected. Subsequently, the "ggvenn" package in R was used to identify the intersecting targets between xylene and NSCLC, which were considered potential intervention targets for xylene in NSCLC. All database searches for this study were conducted up to April 2025.

### Protein-protein interaction (PPI) network construction and Core Targets targets screening

The potential common targets between xylene and NSCLC were imported into the STRING database (Version 12.0) (https://string-db.org/). The species was set as Homo sapiens, and an interaction score of ≥0.4 was used to generate the PPI network diagram, and isolated nodes in the network were hidden. Subsequently, the interaction data output from STRING was imported into Cytoscape software [18] (v3.10.1) to construct a PPI network. Topological analysis of the network nodes was performed using the degree values in the cytoHubba plugin, and potential core targets were selected.

### Enrichment analysis

To explore how xylene might affect NSCLC at the molecular level, we performed GO and KEGG enrichment analyses on the core targets. We used the R packages clusterProfiler, enrichplot and ggplot2 to carry out the analysis. We considered a p.adjust of less than 0.05 to be statistically significant. We then ranked all GO terms and KEGG pathways by Pvalue in ascending order. The top 10 GO terms and the top 30 KEGG pathways were selected. These results helped identify the main biological processes and signaling pathways that may be involved.

## ScRNA-Req analysis

We obtained single-cell RNA sequencing (scRNA-seq) datasets from the Gene Expression Omnibus (GEO) database (GSE198099, https://www.ncbi.nlm.nih.gov/geo/, accessed date April 2025), including samples from two NSCLC patients and two healthy controls. We performed quality control and downstream analyses using the R packages Seurat (v5.1.0). We excluded low-quality cells based on stringent criteria: fewer than 250 and more than 5,000 detected genes, fewer than five total transcripts, or mitochondrial gene content exceeding 5%. After normalization, we selected the 2,000 most highly variable genes for further analysis, processing each sample individually. We integrated the samples using the Integrate-Data function and scaled the data with the ScaleData function. We performed dimensionality reduction by Uniform Manifold Approximation and Projection (UMAP), using the top 15 principal components. Clustering and subclustering analyses were conducted with the FindClusters function at an optimized resolution. Finally, we annotated cell clusters based on the expression profiles of highly expressed genes and established marker genes.

## RNA-seq data sources and differential expression analysis

RNA-seq data for this study were obtained from The Cancer Genome Atlas (TCGA) database (https://portal.gdc.cancer.gov/, accessed date November 2025), comprising 598 lung adenocarcinoma (LUAD) samples, 551 lung squamous cell carcinoma (LUSC) samples, and 108 matched adjacent normal tissue samples. To validate the findings, dataset GSE19188 was additionally downloaded from the GEO database (https://www.ncbi.nlm.nih.gov/geo/, accessed date November 2025), which included 91 non-small cell lung cancer (NSCLC) samples and 65 control samples.

The publicly accessible datasets utilized in this research did not include ethical declarations. Differential expression analysis between groups was performed using the non-parametric Wilcoxon rank-sum test. Visualization of the analysis results was conducted using the ggplot2 package (version 3.5.2). All database searches for this study were conducted up to Novmber 11, 2025.

## Cell culture, RNA extraction, and real-time quantitative PCR

Human lung cancer cell lines (A549, H1299 and PC9) and a human bronchial epithelial cell line (Beas-2B) were purchased from the National Collection of Authenticated Cell Cultures (Shanghai, China; https://www.cellbank.org.cn). All cells were cultured according to the manufacturer's instructions in RPMI-1640 or F-12K medium (HyClone, China) supplemented with 10% fetal bovine serum (Ausbian, Australia) at 37 °C in a humidified 5% $CO_2$ incubator. For RNA extraction, total RNA was isolated from cultured cells using TRIzol reagent (Invitrogen, China). Subsequently, cDNA was synthesized from the extracted RNA using a first-strand cDNA synthesis kit (novoprotein, China). Real-time quantitative PCR (RT-qPCR) was performed on a QuantStudio 3 system (Thermo Fisher Scientific, USA) with SYBR Green Master Mix (Applied Biosystems, USA) to detect the mRNA expression levels of the genes of interest. The relative expression levels were calculated using the $2-\Delta\Delta Ct$ method with GAPDH serving as an internal control. Primers used in RT-qPCR were listed in S3 Table.

## Molecular docking

To explore how xylene interacts with the identified core targets, we carried out molecular docking analysis. This method helped predict both the binding modes and affinities. In this process, we treated the core target proteins as receptors and xylene as the ligand. The 3D structures of the xylene isomers were obtained from the PubChem database. Corresponding protein structures (in PDB format) were downloaded from the PDB database for each core target. CB-Dock2 [19] was used to preprocess the ligands and protein structures. The preprocessing steps included energy minimization, hydrogen addition, removal of water molecules, and structural optimization. After docking, the Vina score was used to evaluate binding affinity. The binding site with the lowest binding energy was selected as the optimal binding mode. We then visualized

the docking results in 3D to better understand the spatial structure of the complexes. Finally, 2D interaction diagrams were generated using Discovery Studio to display key interaction types, such as hydrogen bonds and hydrophobic interactions, between the ligand and the target proteins.

## Statistical analysis

Statistical analysis was performed using R language (version 4.3.3; https://www.R-project.org) and GraphPad Prism 8.0 (GraphPad Software, San Diego, CA, USA). Wilcoxon signed-rank test was employed to compare the expression of core genes mRNA between tumor tissues and normal tissues. One-way analysis of variance (ANOVA) was used for comparisons among multiple groups. A significance level of $p < 0.05$ was considered statistically significant.

## Result

### Identification of potential targets for xylene and NSCLC

We used online databases to predict the potential toxicity of xylene. The toxicity models from ADMETlab 2.0 and ProTox-II indicated that xylene may exhibit toxicological activity associated with carcinogenicity. A total of 259 xylene targets were identified using the ChEMBL and STITCH databases, including 248 from ChEMBL and 13 from STITCH (Fig 1A and S1 Table). Meanwhile, 6,382 targets associated with NSCLC were retrieved from the GeneCards and OMIM databases, including 6,251 from GeneCards and 192 from OMIM (Fig 1B and S2 Table). By intersecting the two gene sets, a total of 115 intersecting targets were obtained that may represent potential targets through which xylene could influence NSCLC (Fig 1C and 1D).

### Analysis of intersecting targets based on the PPI network and identification of core targets

A PPI network was constructed using the STRING database. The resulting PPI network contained 107 nodes and 334 edges. Key targets were subsequently identified by applying median thresholds to three topological measures: Closeness ≥ 0.028, Betweenness ≥ 25.27, Degree ≥ 6.447. Notably, based on node degree values within the network, five core targets were identified: IL1A, H3C13, ITGAM, CCR5, and COMT (Fig 2). These targets may play important roles in cellular processes and disease mechanisms, offering valuable insights into the molecular basis of the involvement of xylene in NSCLC.

### Functional annotation and pathway enrichment analysis of targets

GO and KEGG enrichment analyses were performed on the 107 potential target genes. GO analysis revealed significant enrichment in biological processes such as extracellular matrix (ECM) remodeling, xenobiotic metabolic activation, and lipid-mediated signal transduction (Fig 3A). KEGG pathway analysis further supported the GO findings. The target genes were enriched in pathways including ECM-receptor interaction, focal adhesion, drug metabolism-cytochrome P450, as well as cancer-related pathways such as chemical carcinogenesis-DNA adducts, central carbon metabolism in cancer and the PI3K-Akt signaling pathway (Fig 3B). These findings suggest that the identified targets may play critical roles in the initiation and progression of cancer.

### Single-cell and bulk transcriptomic analysis of hub gene expression in NSCLC

We analyzed four samples from the GSE198099 dataset and obtained approximately 24,469 cells after rigorous quality control. Batch effect correction, dimensionality reduction, and clustering analyses were subsequently performed. Using Uniform Manifold Approximation and Projection (UMAP), we identified 19 distinct cell clusters. These clusters were annotated based on the expression of canonical marker genes, including T&NK cells (CD3D, CD3E, CD3G, NKG7), Epithelial cells (EPCAM, KRT18, KRT19), Macrophages (C1QA, C1QB, C1QC, APOE, APOC1), B cells (CD79A, MS4A1, IGHM,

 

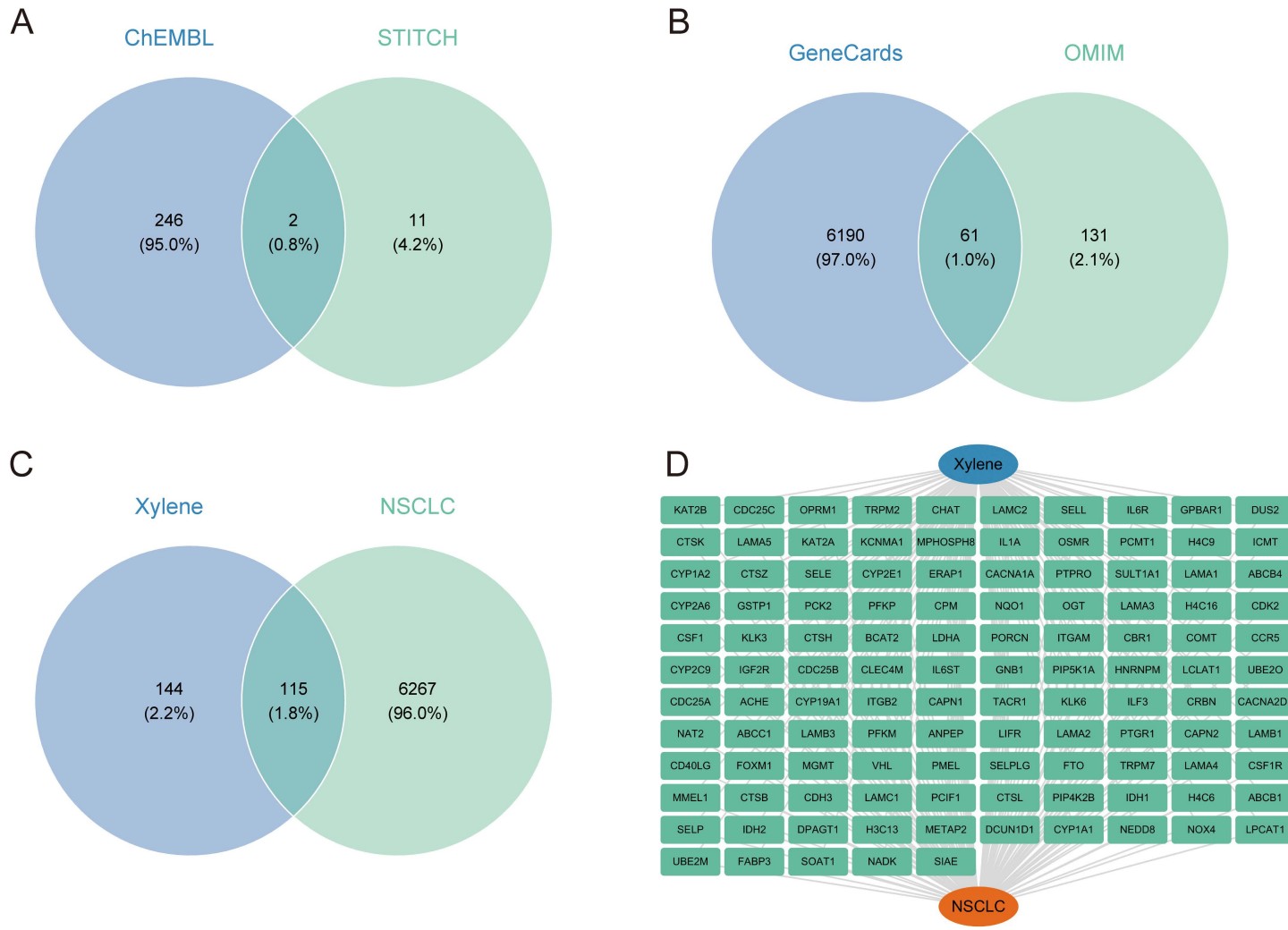

**Fig 1. The associations between xylene and NSCLC. (A)** Venn diagram showing intersecting targets between ChEMBL and STITCH databases **(B)** Venn diagram showing intersecting targets between Genecards and OMIM databases **(C)** Venn diagram showing intersecting targets between xylene and NSCLC **(D)** Relationship Diagram between xylene and NSCLC related genes.

CD79B), Monocytes (CD14, LYZ, FCN1), Endothelial cells (PECAM1, VWF, PLVAP, CD34), Fibroblasts (PDGFRA, FN1, DCN, LUM, ACTA2), and Mast cells (TPSAB1, TPSB2, CPA3) (Fig 4A). The expression patterns of these marker genes across different cell populations are presented in Fig 1B. We further visualized the expression of IL1A, H3C13, ITGAM, CCR5, and COMT. IL1A showed predominant expression in macrophages and monocytes. ITGAM and CCR5 were co-expressed in macrophages, monocytes, and T&NK cells. COMT expression was detected in macrophages, mono-cytes, T&NK cells, and epithelial cells. However, H3C13 expression was not detected in this dataset (Fig 4C).

To systematically validate the expression patterns of the core candidate genes, we conducted a multi-level integrative analysis. Initial transcriptomic analysis of LUAD and LUSC samples from the TCGA database revealed a general upreg-ulation of H3C13 in tumor tissues, whereas IL1A, ITGAM, CCR5, and COMT exhibited a downregulation trend (Fig 5A). Independent validation using the GEO dataset (GSE19188), however, yielded inconsistent findings. Notably, IL1A expres-sion was significantly upregulated, contrasting with the TCGA results, and H3C13 was not detected (Fig 5B). In contrast,

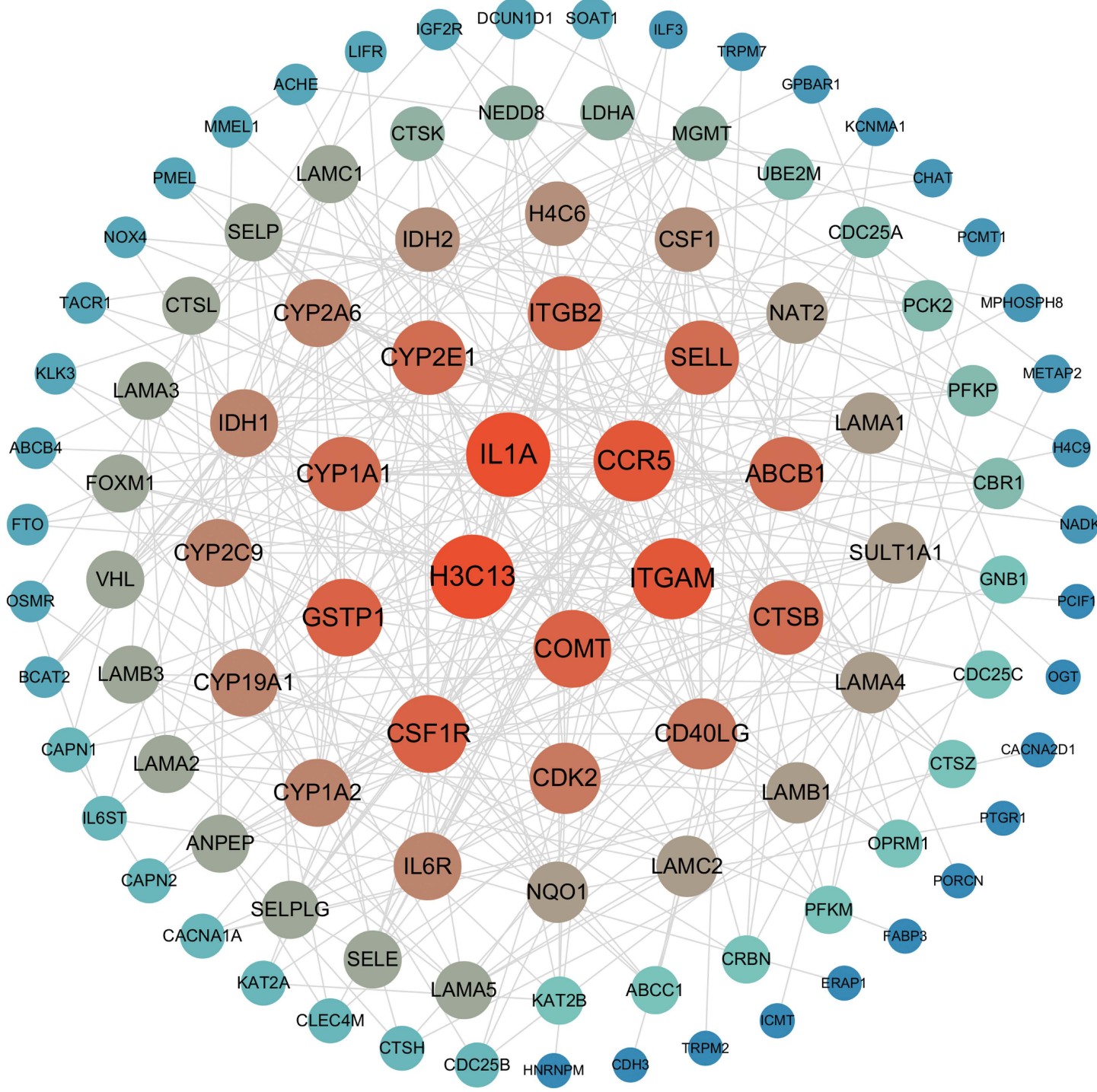

**Fig 2. The PPI network of the potential targets.** An interaction score ≥ 0.4 was applied to construct the PPI network, and isolated nodes were hidden.

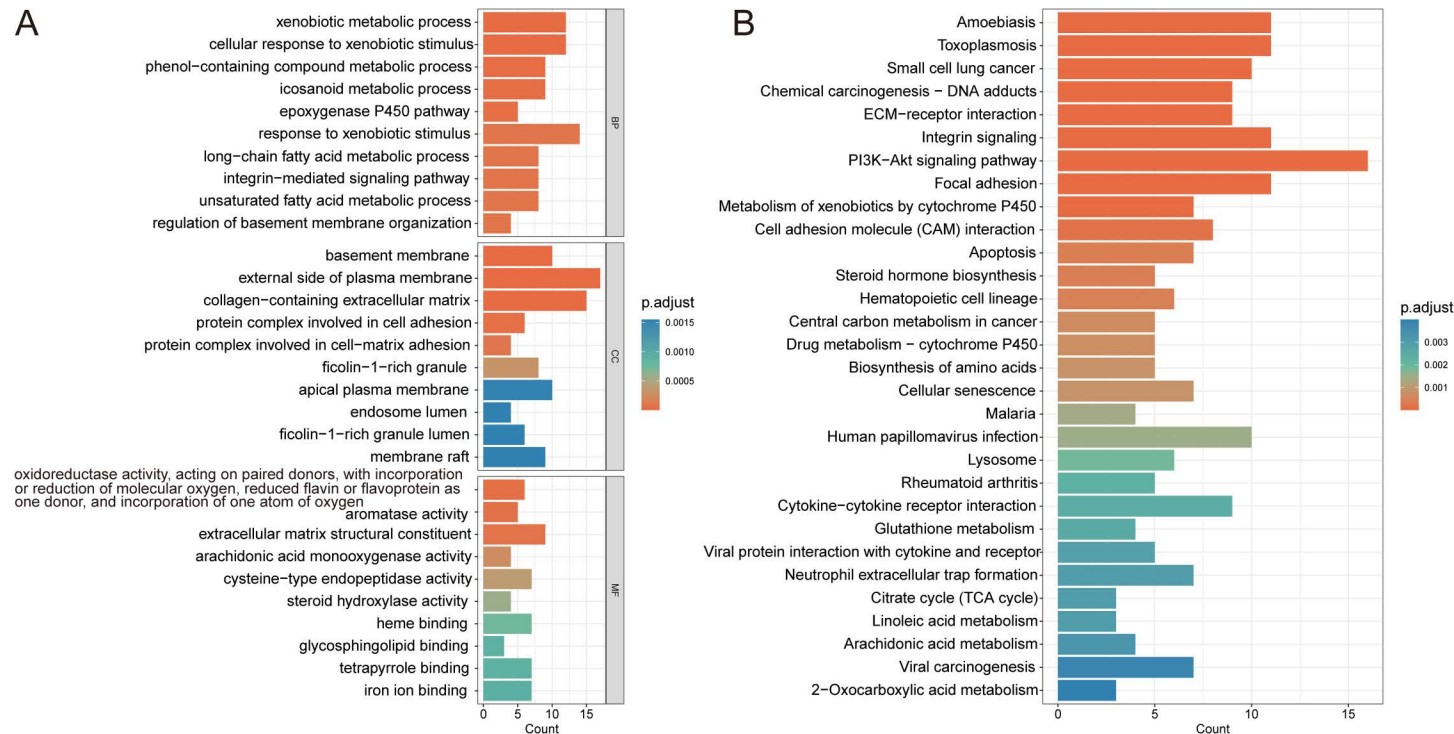

**Fig 3. Enrichment analysis. (A)** GO functional enrichment analysis of potential targets **(B)** KEGG pathway enrichment analysis of potential targets. Terms with an adjusted *p-value* (p.adjust) < 0.05 were considered statistically significant.

the downregulation trends of ITGAM, CCR5, and COMT were consistent with the TCGA analysis (Fig 5B). To address the expression heterogeneity observed between databases, we performed further experimental validation by RT-qPCR in lung cancer cell lines (A549, H1299 and PC9) and normal bronchial epithelial cells (Beas-2B). The in vitro results confirmed the significant downregulation of ITGAM, CCR5, and COMT (Fig 5C). The consistent downregulation trend across both the TCGA and GEO databases suggests that these three genes may represent stable potential targets with reproducible expression patterns. Conversely, although IL1A and H3C13 showed upregulation in the cellular models, their discordant expression profiles in human tissue-derived databases imply that their regulation may be highly dependent on specific cell types or the tumor microenvironment. Therefore, based on their stable and reproducible expression patterns, we identified ITGAM, CCR5, and COMT as the core potential targets linked to xylene exposure.

## Molecular docking between xylene isomers and core targets

Molecular docking analysis was conducted to explore the potential interactions between the three isomers of xylene and the three core targets: ITGAM, CCR5, and COMT. The predicted binding energies from CB-Dock2 for all xylene isomers were below −5.0 kcal/mol. While a binding energy below −5.0 kcal/mol is often considered to suggest potential strong binding affinity in silico, it is important to note that these computational results primarily indicate the theoretical possibility of spontaneous interaction. Lower binding energies generally correspond to more stable predicted complexes. Thus, these preliminary docking analyses suggest that xylene isomers may potentially bind to these core targets, warranting further experimental investigation into their possible role in the molecular mechanisms of xylene-associated NSCLC. The docking poses were visualized in 3D and 2D formats using CB-Dock2 and Discovery Studio, respectively (Figs 6, S1 and S2 and Table 1).

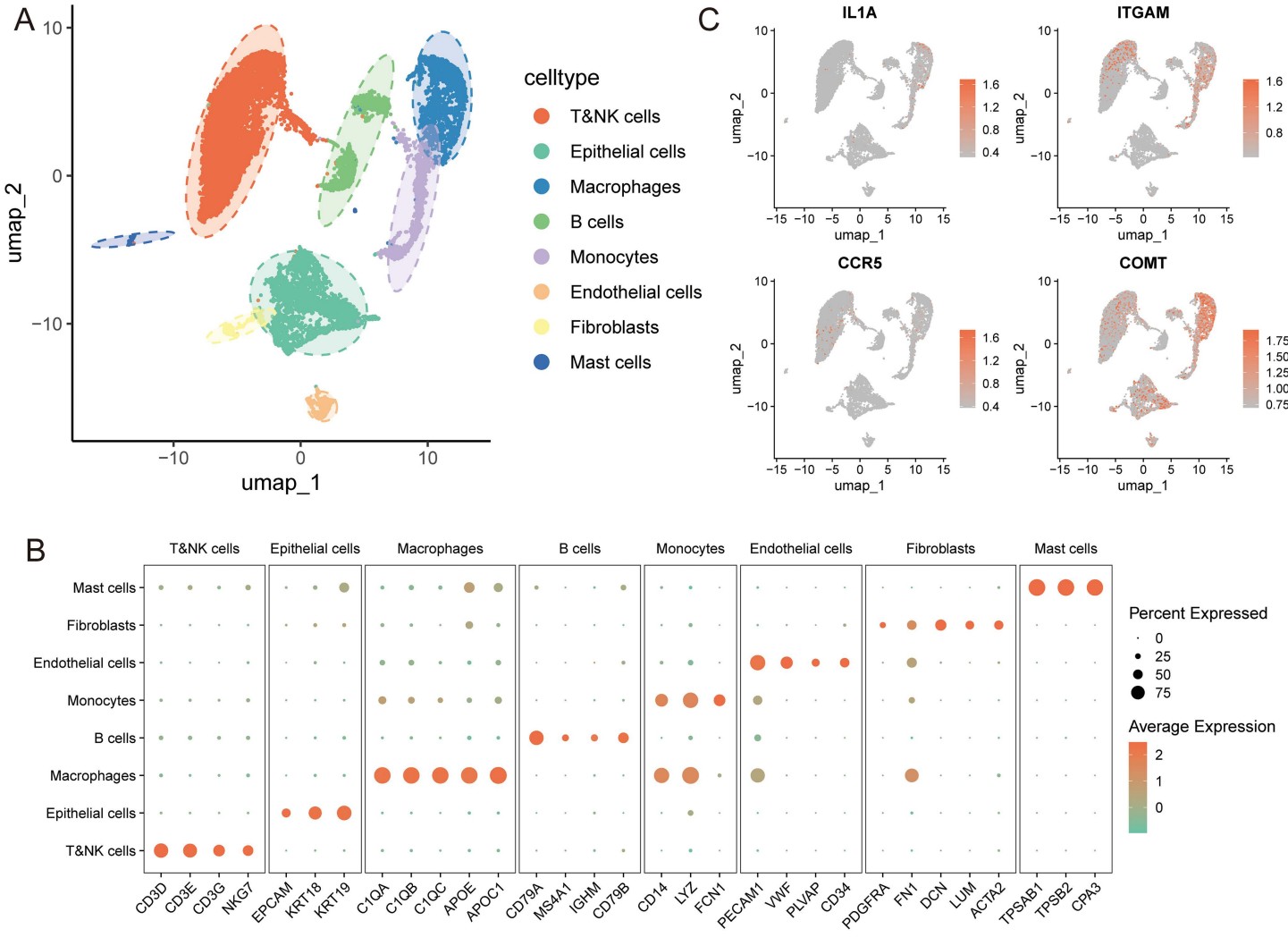

**Fig 4. Results of scRNA-seq analysis. (A)** Cell annotation revealed 8 major cell types **(B)** The association between typical marker genes of different cell types in NSCLC and the 9 major cell types was visually represented through the bubble plo **(C)** The UMAP plot displayed the expression distribution of core targets of different cell types.

## Discussion

Xylene including its ortho, meta and para isomers is a common organic pollutant in the environment due to its widespread use in various commercial products [20]. Although current workplace air concentrations of xylene generally remain within the established occupational exposure limits, studies have shown that prolonged low-dose exposure may still pose health risks to multiple human organ systems [4,21]. This warrants serious attention and highlights the need for strengthened protective and regulatory measures. Epidemiological evidence further suggests a positive association between xylene exposure and increased lung cancer risk. One study reported an OR of 1.44 (95% CI: 1.03–2.01) for lung cancer in relation to xylene exposure, with both ortho- and meta-xylene showing particularly strong associations [8,22]. However, mechanistic studies on xylene inducing lung carcinogenesis remain limited, and the underlying toxicological mechanisms have yet to be fully elucidated.

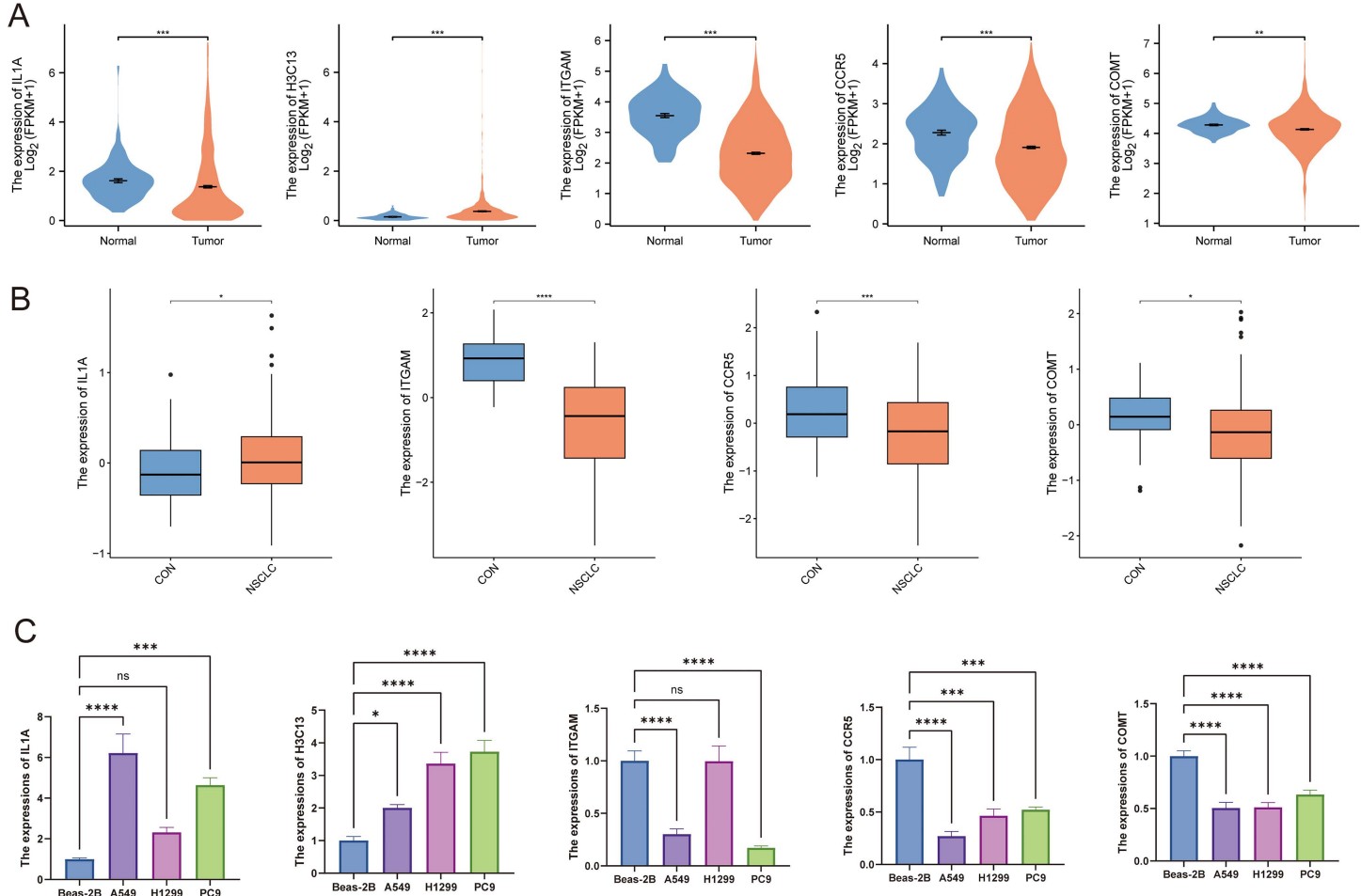

**Fig 5. Integrative validation of core gene expression patterns in lung cancer. (A)** Transcriptomic analysis of candidate genes in NSCLC tissues from the TCGA database, compared with adjacent normal tissues. The box plots show the expression levels. **(B)** Independent validation using the gene expression dataset GSE19188 from the GEO database. **(C)** Experimental validation by RT-qPCR in normal bronchial epithelial cells (Beas-2B) and lung cancer cell lines (A549, H1299, PC9). ns, $p > 0.05$, * $p < 0.05$, ** $p < 0.01$, *** $p < 0.001$.

This study conducted a network toxicology analysis using data from the ChEMBL, STITCH, GeneCards, and OMIM databases to identify potential targets associated with xylene inducing NSCLC. A PPI network was constructed through the STRING platform and visualized with Cytoscape software, leading to the identification of five core targets: IL1A, H3C13, ITGAM, CCR5, and COMT. GO functional enrichment and KEGG pathway analyses were performed to explore the biological functions and signaling pathways related to these targets. To further elucidate their cellular relevance, we examined the expression patterns of the core targets across diverse cell subpopulations using scRNA-seq data. Multi-platform validation based on TCGA, GEO, and cellular experiments revealed that ITGAM, CCR5, and COMT displayed consistent and reproducible downregulation in NSCLC, whereas IL1A and H3C13 showed inconsistent expression across datasets. Following this stringent evaluation of expression concordance, ITGAM, CCR5, and COMT were ultimately identified as the most robust targets potentially linked to xylene exposure. Furthermore, molecular docking was carried out via PubChem, CB-Dock2, and Discovery Studio to predict the potential intermolecular interactions between xylene and the core target proteins. In summary, this study reveals potential molecular mechanisms by which xylene may contribute

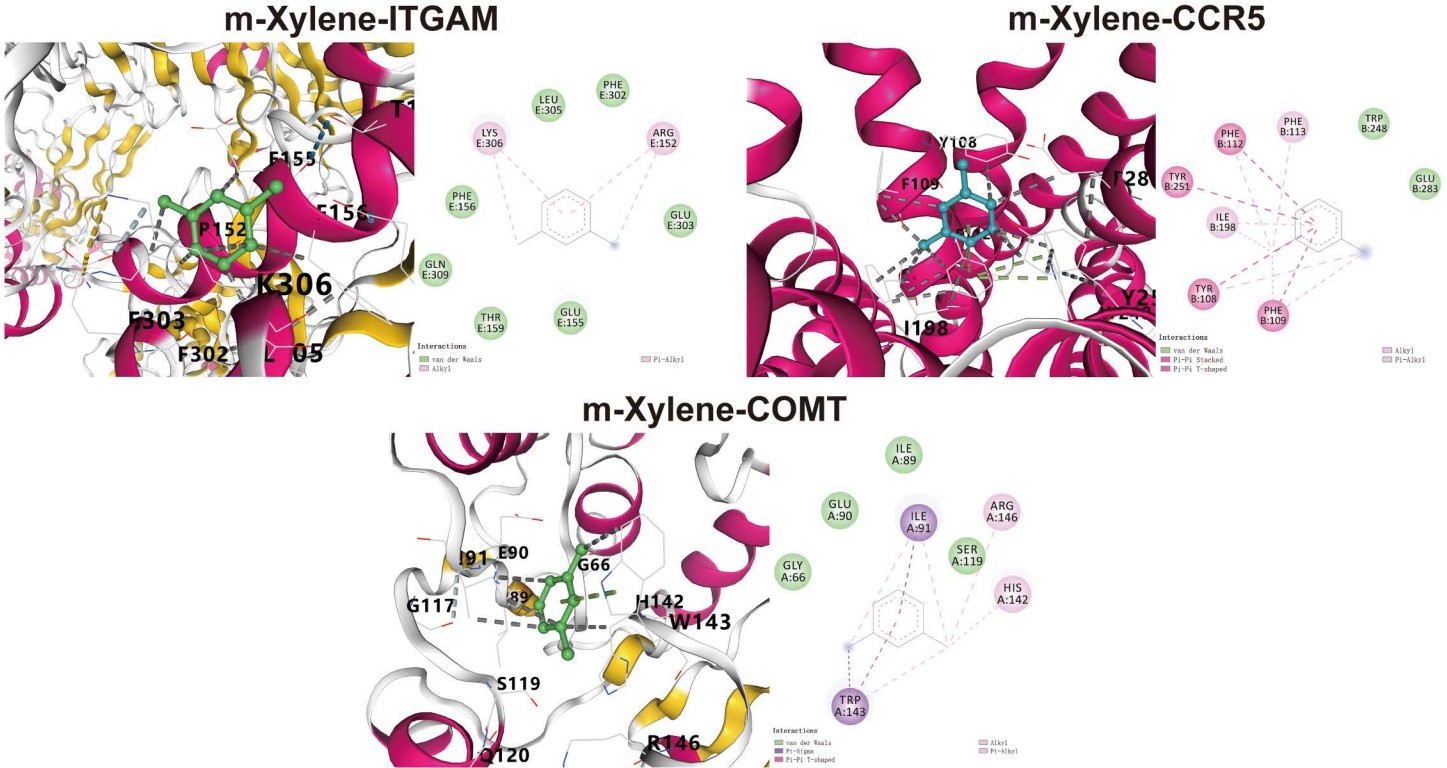

**Fig 6. Molecular docking results of m-xylene with core targets, displaying the lowest binding energies in both 3D and 2D formats.**

**Table 1. Binding energy for target with xylene.**

| Target | Degree | PDB number | Hydrone | Binding energy (kcal/mol) |
|---|---|---|---|---|
| ITGAM | 6 | 3Q3G | m-Xylene | −5.9 |
| | | | o-Xylene | −5.7 |
| | | | p-Xylene | −5.3 |
| CCR5 | 15 | 4MBS | m-Xylene | −6.1 |
| | | | o-Xylene | −5.8 |
| | | | p-Xylene | −5.5 |
| COMT | 12 | 3A7E | m-Xylene | −6.3 |
| | | | o-Xylene | −6.1 |
| | | | p-Xylene | −6 |

to NSCLC development, providing a theoretical basis for understanding its toxicological effects and informing disease prevention efforts.

ITGAM (CD11b) is an important myeloid integrin predominantly expressed on macrophages and monocytes, where it participates in adhesion, phagocytosis, inflammatory regulation, and innate immune responses [23,24]. Consistent with these functions, our scRNA-seq data show clear ITGAM expression in macrophages, monocytes, and a subset of T and NK cells. Accumulating evidence indicates that the role of ITGAM varies across cell types and immune microenvironments. It is often upregulated in immunosuppressive tumor-associated macrophages and contributes to a suppressed

immune state [24,25]. ITGAM also influences dendritic cell antigen presentation and T cell activation [26], affects the cytotoxic activity of NK cells, and has been linked to immune evasion and tumor progression mediated by myeloid-derived suppressor cells [27,28]. Together, these observations suggest that ITGAM may exert multiple layers of immunoregulatory activity in the microenvironment of NSCLC. In this study, molecular docking suggested a potential interaction between xylene and the ITGAM protein. Although this finding does not indicate a direct causal relationship, it raises the possibility that environmental exposure could influence signaling pathways in myeloid cells and modify the immune landscape of lung cancer. Further experimental studies are needed to clarify this potential mechanism.

CCR5, a G protein-coupled receptor for chemokines such as CCL5, is a central mediator of immune cell trafficking into the tumor microenvironment, and its function in NSCLC appears to be highly context dependent. Activation of the CCR5/ CCL5 axis has been shown to recruit immunosuppressive M2-like tumor-associated macrophages, thereby fostering an immunosuppressive microenvironment and promoting tumor progression [29]. However, CCR5 activity varies across immune landscapes. In lung adenocarcinoma, high CCR5 expression correlates with increased infiltration of M0 macrophages and memory B cells, a pattern that may enhance responsiveness to immunotherapy [30]. Genetic studies further indicate that CCR5 links host susceptibility to the tumor immune milieu: functional promoter polymorphisms such as rs1799987 modulate transcriptional activity and influence NSCLC risk, while high-risk haplotypes such as H5 are associated with both increased susceptibility and distinct immune-infiltration profiles [31]. Based on these findings, we hypothesize that xylene exposure may alter CCR5-related signaling or structural properties and, together with CCR5 promoter polymorphisms, modulate immune-cell recruitment, ultimately disrupting immune homeostasis within the lung cancer microenvironment.

COMT is an enzyme that catalyzes the methylation of catechol substrates and plays important roles in various diseases, including cardiovascular disorders and cancer [32]. The COMT rs4680 polymorphism has been significantly associated with NSCLC susceptibility [33]. Moreover, COMT may influence tumor progression by regulating estrogen metabolism [34]. The Val158Met polymorphism reduces catecholamine metabolism efficiency, leading to the accumulation of DNA oxidative damage, which markedly increases NSCLC risk in female smokers [34]. These findings imply that xylene exposure may exacerbate genetic susceptibility in certain populations.

The results of GO and KEGG enrichment analyses suggest that xylene may contribute to the development and progression of NSCLC through multiple key biological processes and signaling pathways. The associated targets were mainly enriched in biological processes such as ECM remodeling, xenobiotic metabolic activation, and lipid-mediated signal transduction. These genes were also involved in typical cancer-related pathways, including ECM–receptor interaction, focal adhesion, and the PI3K-Akt signaling pathway. ECM remodeling is recognized as a hallmark of tumor progression. Studies have shown that the ECM enhances tumor cell invasion and metastasis through collagen cross-linking, proteolytic degradation, and mechanical remodeling [35]. In NSCLC, CD248-positive cancer-associated fibroblasts (CAFs) can activate the Hippo pathway to induce CTGF expression, which promotes collagen I deposition in the stroma [36]. This process increases ECM stiffness and consequently drives tumor metastasis. Beyond serving as a structural scaffold, the ECM also functions as a signaling platform. It regulates the tumor microenvironment by storing and releasing cytokines, chemokines, and growth factors, thereby influencing NSCLC progression [37]. ECM–receptor interaction activates PI3K through integrin signaling, leading to Akt phosphorylation, which in turn suppresses apoptosis and stimulates mTOR signaling to promote cell proliferation and angiogenesis [38]. Dysregulation of the PI3K/Akt/mTOR pathway is a major oncogenic driver in NSCLC, facilitating tumor progression by enhancing cell proliferation, inhibiting apoptosis, reprogramming metabolism, and increasing drug resistance [39]. In chemical carcinogenesis pathways, xenobiotic compounds can undergo metabolic activation to form DNA adducts, which may lead to proto-oncogene mutations and activate oncogenic signals such as the PI3K-Akt pathway [40]. This cross-talk between multiple signaling pathways forms a positive feedback loop that accelerates NSCLC progression. Furthermore, enrichment in xenobiotic metabolic activation suggests that some target genes may be involved in the metabolism and detoxification of carcinogens [41]. Among them, the cytochrome

P450 (CYP450) family plays a central role in the metabolic activation of exogenous chemicals. While this process is essential for detoxification, it may also result in the formation of DNA adducts, thereby increasing mutagenic risk [42]. In summary, xylene may contribute to NSCLC initiation and progression by modulating ECM function and activating the PI3K-Akt signaling axis. This mechanism involves both carcinogen metabolism and the coordinated regulation of multiple cancer-related pathways, providing a theoretical foundation for further investigation into its carcinogenic potential and strategies for prevention.

Molecular docking analysis indicated that the five core target proteins play a critical role in xylene-induced NSCLC. These proteins formed stable interactions with the active sites of the three structural isomers of xylene, with binding energies ranging from –6.3 to –4.8 kcal/mol, suggesting strong binding affinities.

This study has certain limitations. First, there is a lack of in vivo experimental evidence to validate the specific pathogenic effects of xylene on NSCLC. Therefore, the biological relevance of the predicted core targets and their associated signaling pathways in tumor development cannot be confirmed at the molecular level. Future studies should investigate the molecular mechanisms through animal models or cell experiments to facilitate the translation and application of key findings. Secondly, NSCLC includes various subtypes such as adenocarcinoma, squamous cell carcinoma, and large cell carcinoma, each with distinct molecular characteristics and pathogenic mechanisms. This study did not analyze the subtypes in detail, which may obscure subtype-specific target distribution and carcinogenic pathways, potentially affecting the accuracy and clinical relevance of the findings. Additionally, since this study did not conduct a dose-response analysis based on specific occupational or environmental exposure levels, there are limitations in evaluating the health risk intensity under current exposure conditions. Incorporating real-world data into the risk assessment framework could enhance the practical guidance of the findings for occupational health management and policy development. Finally, as with any database-driven approach, the identification of intersecting targets may include false-positive associations due to inherent database biases and incomplete annotations. Although confidence filters were applied, these results should be interpreted as computational predictions requiring experimental confirmation.

## Conclusion

This study systematically identifies ITGAM, CCR5, and COMT as putative molecular links between xylene exposure and NSCLC pathogenesis. Beyond providing mechanistic hypotheses, our findings may offer several translational directions for future investigation. First, these genes could be further evaluated as candidate biomarkers for risk stratification in populations with occupational exposure. Second, our results suggest that exposure guidelines might ultimately incorporate inter-individual genetic susceptibility, including variation in COMT-related pathways. Third, the conceptual possibility of repurposing CCR5 antagonists or developing metabolic modulators warrants exploration in future preclinical models. Collectively, this work provides a hypothesis-generating framework that may guide the development of targeted strategies for mitigating xylene-associated lung cancer risk.

## Supporting information

**S1 Fig. Molecular docking results of o-xylene with core targets.**
(TIF)

**S2 Fig. Molecular docking results of p-xylene with core targets.**
(TIF)

**S1 Table. Xylene Targets Identified in ChEMBL and STITCH Databases.**
(XLS)

**S2 Table. NSCLC Targets in GeneCards and OMIM.**
(XLS)

**S3 Table. Primer for RT-qPCR.**
(XLS)

## Author contributions

**Conceptualization:** Yuanlin Qi.

**Data curation:** Mingfang Zhang, Yuanlin Qi.

**Formal analysis:** Hongquan Chen, Xiangpeng Chen.

**Funding acquisition:** Weibin Lin, Mingfang Zhang, Yuanlin Qi.

**Investigation:** Xiangpeng Chen, Renxi Lin.

**Methodology:** Hongquan Chen, Xiangpeng Chen, Qing Chen.

**Resources:** Weibin Lin, Yuanlin Qi.

**Software:** Xiangpeng Chen.

**Supervision:** Xiangpeng Chen.

**Validation:** Qing Chen.

**Visualization:** Hongquan Chen, Xiangpeng Chen.

**Writing – original draft:** Hongquan Chen.

**Writing – review & editing:** Mingfang Zhang, Yuanlin Qi.

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
