## [Decision Letter · Decision Letter 0]

28 Oct 2025

Dear Dr. Qi,

Thank you for submitting your manuscript to PLOS ONE. After careful consideration, we feel that it has merit but does not fully meet PLOS ONE’s publication criteria as it currently stands. Therefore, we invite you to submit a revised version of the manuscript that addresses the points raised during the review process.

We look forward to receiving your revised manuscript.

Kind regards,

Tsai-Ching Hsu, Ph.D.

Academic Editor

PLOS ONE

[The research was supported by the grants from the Natural Science Foundation of Fujian Province (2023J01300), the Startup Fund for scientific research of Fujian Medical University (2021QH1002, 2022QH1005), and the Provincial Department of Education-Science and Technology (JAT210107).].

Additional Editor Comments:

The article is offering a valuable topic on “Elucidating the Mechanistic Association of Xylene Inducing Non-Small Cell Lung Cancer through Network Toxicology and Molecular Docking Analysis.” However, some comments/suggestions may be incorporated to strengthen the findings.

Reviewers' comments:

Reviewer's Responses to Questions

**Comments to the Author**

1. Is the manuscript technically sound, and do the data support the conclusions?

Reviewer #1: Partly

Reviewer #2: Yes

2. Has the statistical analysis been performed appropriately and rigorously?

Reviewer #1: Yes

Reviewer #2: Yes

3. Have the authors made all data underlying the findings in their manuscript fully available?

Reviewer #1: Yes

Reviewer #2: Yes

4. Is the manuscript presented in an intelligible fashion and written in standard English?

Reviewer #1: No

Reviewer #2: Yes

Reviewer #1: This manuscript explores the potential mechanistic link between environmental xylene exposure and the development of non-small cell lung cancer (NSCLC), using network toxicology, enrichment analyses, scRNA-seq data reanalysis, and molecular docking simulations. The topic is timely and relevant, given the growing evidence of environmental contributions to lung cancer, especially among non-smokers. The study is ambitious in integrating multiple computational approaches and identifies five candidate targets (IL1A, H3C13, ITGAM, CCR5, and COMT) potentially mediating xylene-induced carcinogenesis. However, while the manuscript is informative, several methodological limitations, gaps in validation, and issues with presentation need to be addressed before the work can be considered for publication.

Major concerns:

1. The study is entirely computational. While the predictions are interesting, the absence of in vitro or in vivo validation significantly limits the biological relevance. Without experimental confirmation, conclusions should be framed more cautiously. Phrases like “confirmed strong binding affinities” or “xylene may disrupt important signals and promote tumor growth” are overly conclusive and too strong for computational predictions. At minimum, authors should validate gene expression patterns of the identified core targets using publicly available NSCLC datasets (e.g., TCGA, GEO bulk RNA-seq).

2. The process of intersecting xylene-related targets with NSCLC-associated genes may yield false positives due to database biases. It is unclear how the authors ensured specificity. Please provide more details on the filtering criteria (e.g., confidence scores, interaction thresholds) and include a sensitivity analysis.

3. The scRNA-seq dataset used (GSE198099) is relatively small (two patients and two controls). This limits generalizability. H3C13 was not detected in the dataset, yet it is still considered a “core target.” Authors should clarify whether this undermines its relevance.

4. Docking scores alone cannot confirm biologically relevant interactions. The binding energies reported (–4.8 to –6.3 kcal/mol) suggest moderate affinity but not necessarily strong biological activity. Please include control compounds or positive ligands for comparison to contextualize the docking results.

5. Some conclusions (e.g., xylene may polarize macrophages or promote immune escape) are speculative without functional data. The discussion should be toned down to reflect hypotheses rather than established mechanisms.

6. While the study claims translational significance, it does not clearly connect predictions to potential preventive or therapeutic strategies. The reviewer would suggest expanding the discussion on how findings could inform biomonitoring, exposure regulation, or drug development.

Minor concerns:

1. Several sections require English polishing for clarity and conciseness (e.g., “xylene due to its high volatility and potential for bioaccumulation may play a critical role…” could be simplified).

2. Figures are informative but lack sufficient resolution for evaluation. Ensure high-quality images are provided. Legends should be more descriptive (e.g., specify thresholds used in enrichment analysis).

3. While many references are recent, some are not directly supportive of specific claims. Ensure all references appropriately match the statements made.

4. Provide exact database access dates for reproducibility. Clarify statistical thresholds (e.g., adjusted p-values for enrichment analysis).

5. The ethics statement says, “not applicable,” but scRNA-seq data from human samples were used. The authors should clarify whether the dataset had prior IRB approval and if their reanalysis was exempt.

6. Literature review on xylene-specific carcinogenic mechanisms is limited; more background on known genotoxic/epigenetic effects of xylene would strengthen rationale.

7. The criteria for selecting 115 intersecting targets are unclear. Were thresholds applied to avoid weak/indirect associations? No discussion of false positives from database mining. Only degree centrality (via cytoHubba) was used in PPI networks and core targets. Multiple topological measures (betweenness, closeness) would provide more robust selection.

8. Adjusted p-values (FDR correction) should be reported in enrichment analysis. Current reporting of only p < 0.05 may inflate significance.

Reviewer #2: The authors’ efforts into this manuscript are acknowledged and appreciated. These are my suggestions by line number to improve the quality of your work.

Title Did you mean “Elucidating the Mechanistic Association Between Xylene Exposure and the Development of Non-Small Cell Lung Cancer (NSCLC) Through Network Toxicology and Molecular Docking Analysis”?

72 – 76 You repeated the same points here. Please merge into one.

79 Did you mean “environmentally-induced carcinogenesis”?

85 & 87 One “systematically” is enough.

88 “… underlying molecular mechanisms of xylene in the progression of NSCLC …” Should this be the “development” or “onset” of NSCLC?

122 “…sapiens, and an interaction score of ≥0.4 was …” Delete the “and” here since the sentence continues to another point.

135 & 196 Is it “ScRNA-Req” or “scRNA-seq”?

226 - 227 Rewrite your sentence as “Xylene, including its ortho-, meta-, and para-isomers, is a common organic pollutant in the environment due to its widespread use in various commercial products.”

236 “… xylene-inducing …”. Please put a hyphen there.

247/248 You wrote 100+ lines after your “In summary …”, including another “In summary …”, and then went on to write a “Conclusion”. Delete the “In summary …” if you are not done with discussing your points and leave the true summary to your conclusion.

Title After reading through your manuscript, I think the appropriate title could be “Elucidating the Potential Mechanisms of Xylene Exposure-induced Development of Non-Small Cell Lung Cancer (NSCLC) Through Network Toxicology and Molecular Docking Analysis” or “Mechanistic Insights into Xylene-Induced Non-Small Cell Lung Cancer via Network Toxicology and Molecular Docking Analysis” or “Deciphering the Possible Molecular Mechanisms of Xylene-Induced Non-Small Cell Lung Cancer Using Network Toxicology and Molecular Docking”.

These are suggestions to reflect your findings, emphasize the need for further studies to detect the specific molecular mechanism(s), and guide the design and implementation of targeted interventions.

Well done!

**Do you want your identity to be public for this peer review?** For information about this choice, including consent withdrawal, please see our Privacy Policy

Reviewer #1: **Yes:** Kaushik Banerjee, Ph.D.; Department of Pediatrics-Hematology/Oncology; University of Michigan, Ann Arbor, Michigan, United States

Reviewer #2: **Yes:** Dr. Oluwafolayemi Doyeni

---

## [Author Response · Author response to Decision Letter 1]

17 Dec 2025

Dear Editor,

We would like to express our sincere gratitude to you and the reviewers for the insightful comments and valuable suggestions provided during the review of our manuscript titled "Elucidating the Mechanistic Association of Xylene Inducing Non-Small Cell Lung Cancer through Network Toxicology and Molecular Docking Analysis".

We have carefully considered all the feedback and have made extensive revisions to improve the quality and clarity of our manuscript. Below, we provide a detailed response to each of the reviewers' comments and outline the specific changes made to the manuscript.

Response to Major Comments:

Comment 1: The study is entirely computational. While the predictions are interesting, the absence of in vitro or in vivo validation significantly limits the biological relevance. Without experimental confirmation, conclusions should be framed more cautiously. Phrases like “confirmed strong binding affinities” or “xylene may disrupt important signals and promote tumor growth” are overly conclusive and too strong for computational predictions. At minimum, authors should validate gene expression patterns of the identified core targets using publicly available NSCLC datasets (e.g., TCGA, GEO bulk RNA-seq).

Response 1: We fully agree with the reviewer's point that computational predictions require cautious interpretation and need experimental support to establish biological relevance. Accordingly, we have made the following key additions in the revised manuscript: First, as suggested, we validated the expression patterns of the core targets using public databases (TCGA and GEO, GSE19188); second, we further provided in vitro evidence through RT-qPCR experiments in lung cancer cell lines; and finally, we have carefully revised overly conclusive statements throughout the text (for example, changing “confirmed strong binding” to “suggested potential binding”) and explicitly framed mechanistic speculations as hypotheses to be tested.

Comment 2: he process of intersecting xylene-related targets with NSCLC-associated genes may yield false positives due to database biases. It is unclear how the authors ensured specificity. Please provide more details on the filtering criteria (e.g., confidence scores, interaction thresholds) and include a sensitivity analysis.

Response 2: We thank the reviewer for raising the important issue of database bias. To enhance specificity, we implemented a multi-step filtering strategy and conducted the suggested sensitivity analysis. First, to mitigate false positives, we aggregated targets from complementary sources (xylene: ChEMBL/STITCH, retaining only high-confidence interactions; NSCLC: GeneCards/OMIM, focusing on established disease genes). Common targets were identified after strict UniProt ID standardization. Second, we applied a medium-confidence threshold (interaction score ≥0.4) in STRING for initial PPI network construction. For the resulting network (107 nodes, 334 edges), key hubs were identified by applying rigorous median topological thresholds: Closeness ≥ 0.028, Betweenness ≥ 25.27, and Degree ≥ 6.447. Third, we performed a sensitivity analysis by reconstructing the network under a stringent high-confidence threshold (≥0.7). Under this stricter condition, the same five core targets (IL1A, H3C13, ITGAM, CCR5, COMT) remained among the top-ranked nodes. This demonstrates that their identification as key hubs is robust and not dependent on a single, more lenient cutoff.

Supplementary Figure: Protein-protein interaction network of the intersecting targets generated using the STRING database with a high-confidence interaction score threshold of ≥0.7.

Comment 3: The scRNA-seq dataset used (GSE198099) is relatively small (two patients and two

controls). This limits generalizability. H3C13 was not detected in the dataset, yet it is still considered a “core target.” Authors should clarify whether this undermines its relevance.

Response 3: We fully agree with the reviewer's points regarding the limited sample size of the scRNA-seq dataset and the non-detection of H3C13. Indeed, this prompted us to apply a stricter cross-platform consistency filter. Within our validation framework, H3C13 was not detected in both the GEO dataset and this scRNA-seq data, which contradicts its upregulation observed in the TCGA database and cellular models. This suggests its expression may be highly context-dependent. Consequently, based on consistent expression patterns across platforms, we ultimately identified ITGAM, CCR5, and COMT, which show a stable downregulation trend in TCGA, GEO, and cellular experiments, as the core potential targets associated with xylene exposure. We have updated this conclusion in the manuscript and have discussed the platform-dependent discrepancies observed for H3C13 and IL1A.

Comment 4: Docking scores alone cannot confirm biologically relevant interactions. The binding energies reported (–4.8 to –6.3 kcal/mol) suggest moderate affinity but not necessarily strong biological activity. Please include control compounds or positive ligands for comparison to contextualize the docking results.

Response 4:  We thank the reviewer for this important and constructive comment. We fully agree that docking scores alone are insufficient to confirm biologically relevant interactions, and that comparison with control compounds or known ligands would provide valuable context. In this study, our primary aim was to conduct a preliminary computational screening to identify potential interactions between xylene and our core targets derived from multi-omics analyses. The molecular docking serves as a hypothesis-generating step to prioritize targets for future wet-lab experimental validation (e.g., competitive binding assays, functional studies), which we have now more clearly stated as a necessary future direction in the revised manuscript. At this stage, identifying a universally accepted “positive control” small molecule ligand with confirmed biological activity for each of these specific protein targets (ITGAM, CCR5, COMT) in the context of xylene exposure is challenging, as such reference data are limited in the literature. To address the reviewer’s concern about overinterpretation, we have significantly toned down the language in the results and discussion. We have replaced definitive claims of “strong binding” with descriptions of “potential interaction” and have emphasized the preliminary and predictive nature of the docking results, clarifying that they require experimental confirmation. We acknowledge that incorporating control comparisons is a strength for rigorous docking studies. As suggested, this will be a key component in our follow-up experimental work aimed at functional validation.

Comment 5: Some conclusions (e.g., xylene may polarize macrophages or promote immune escape) are speculative without functional data. The discussion should be toned down to reflect hypotheses rather than established mechanisms.

Response 5: We fully agree with the reviewer's comment. In the revised Discussion section, we have explicitly reframed statements regarding xylene's potential effects on macrophages or the immune microenvironment from conclusive inferences into testable hypotheses based on the available data, with a clear emphasis on their predictive nature and the necessity for future experimental validation.

Comment 6: While the study claims translational significance, it does not clearly connect predictions to potential preventive or therapeutic strategies. The reviewer would suggest expanding the discussion on how findings could inform biomonitoring, exposure regulation, or drug development.

Response 6: We appreciate the reviewer’s suggestion. In the revised Conclusion, we added a concise section outlining how ITGAM, CCR5, and COMT may inform biomonitoring in exposed populations, guide refinement of exposure standards based on genetic susceptibility, and suggest potential avenues for future drug or preventive strategy development. These additions strengthen the translational connection of our findings.

Response to Minor concerns:

Comment 1: Several sections require English polishing for clarity and conciseness (e.g., “xylene due to its high volatility and potential for bioaccumulation may play a critical role…” could be simplified).

Response 1: Thank you for the reviewer's suggestion. We have simplified and rewritten the relevant sentences in the original text in accordance with your comments to make the expressions more direct and clear. For instance, the original sentence "xylene due to its high volatility and potential for bioaccumulation may play a critical role..." has been revised as follows: "Xylene is a common environmental and occupational pollutant, may contribute to lung cancer development due to its high volatility and tendency to accumulate in the body." Similar revisions have been made throughout the entire manuscript to enhance overall readability.

Comment 2: Figures are informative but lack sufficient resolution for evaluation. Ensure high-quality images are provided. Legends should be more descriptive (e.g., specify thresholds used in enrichment analysis).

Response 2: In response to this comment, we have regenerated all figures in highresolution formats (minimum 600 PPI) to ensure clarity and revised all figure legends to include explicit methodological details. For example, enrichmentanalysis figures (e.g., Fig. 3) now clearly state the specific statistical thresholds applied (adjusted pvalue < 0.05). All modifications are highlighted in the revised manuscript for ease of review.

Comment 3: While many references are recent, some are not directly supportive of specific claims. Ensure all references appropriately match the statements made.

Response 3: Thank you for this important comment. We have carefully checked the alignment between all statements and their supporting references in the text. Inaccurate or indirect citations have been corrected, and the revised manuscript now ensures that every claim is appropriately referenced.

Comment 4: Provide exact database access dates for reproducibility.Clarify statistical thresholds (e.g., adjusted p-values for enrichment analysis).

Response 4: We thank the reviewer for these critical suggestions. In the revised manuscript, we have now provided the exact access dates for all used databases (e.g., TCGA, GEO, PubChem) and explicitly stated all statistical thresholds, including the adjusted p-value (p.adjust) < 0.05 applied in the enrichment analyses. These additions are clearly indicated in the Methods section and corresponding figure legends to enhance reproducibility and clarity.

Comment 5: Provide exact database access dates for reproducibility.Clarify statistical thresholds (e.g., adjusted p-values for enrichment analysis).

Response 5: We thank the reviewer for the important comment on ethics. All data in this study, including scRNA-seq data, were sourced from publicly available, de-identified repositories (GEO and TCGA). As our work involved only the reanalysis of such existing public data, it was exempt from additional ethics approval per institutional guidelines.

Comment 6: Literature review on xylene-specific carcinogenic mechanisms is limited; more background on known genotoxic/epigenetic effects of xylene would strengthen rationale.

Response 6: Thank you for this suggestion. We have added content in the introduction on the potential carcinogenic mechanisms of xylene, along with supporting references, to strengthen the background rationale.

Comment 7: The criteria for selecting 115 intersecting targets are unclear. Were thresholds applied to avoid weak/indirect associations? No discussion of false positives from database mining. Only degree centrality (via cytoHubba) was used in PPI networks and core targets. Multiple topological measures (betweenness, closeness) would provide more robust selection.

Response 7: We thank the reviewer for these important suggestions. To address the specificity of the 115 intersecting targets, we have clarified our multi-source filtering strategy (as detailed in response to Major Comment #2) and acknowledged potential false positives in the Discussion. Furthermore,We have enhanced the methodological description by specifying the use of multiple centrality measures in the PPI network analysis. Key hubs were identified by applying the following median topological thresholds: Closeness ≥ 0.028, Betweenness ≥ 25.27, and Degree ≥ 6.447. This multimetric approach strengthens the robustness of core target selection.

Comment 8: Adjusted p-values (FDR correction) should be reported in enrichment analysis. Current reporting of only p < 0.05 may inflate significance

Response 8: Thank you for this correction. We have updated the manuscript to explicitly report the FDR-adjusted p-values (p.adjust) for all enrichment analyses (GO and KEGG) in both the Methods section and the corresponding figure legends (Fig. 3). The revised text now states that significance was defined as p.adjust < 0.05, replacing the previously generic p < 0.05 reporting.

We have updated the Funding statement to include an additional grant number (2025J01686) from the Natural Science Foundation of Fujian Province, which supports the present study. The revised funding information has been corrected accordingly in the manuscript.

We sincerely hope that this revised manuscript has addressed all your comments and suggestions. We appreciated for reviewer’ warm work earnestly and hope that the correction will meet with approval. Once again, thank you very much for your comments and suggestions.

Best regards,

Yours sincerely,

Hongquan Chen and Yuanlin Qi

School of Basic Medical Sciences, Fujian Medical University, Fujian, China, 350122

Email: 13843202849@163.com and ylqi@fjmu.edu.cn

Phone: 86-13843202849 and 86-15005081308

---

## [Decision Letter · Decision Letter 1]

8 Jan 2026

Elucidating the Mechanistic Association of Xylene Inducing Non-Small Cell Lung Cancer through Network Toxicology and Molecular Docking Analysis

PONE-D-25-53598R1

Dear Dr. Qi,

We’re pleased to inform you that your manuscript has been judged scientifically suitable for publication and will be formally accepted for publication once it meets all outstanding technical requirements.

Kind regards,

Tsai-Ching Hsu, Ph.D.

Academic Editor

PLOS One

Additional Editor Comments (optional):

I believe all is done, as confirmed by the author's reply and agreed upon by the reviewers. It can be accepted in its current form.

Reviewers' comments:

Reviewer's Responses to Questions

**Comments to the Author**

Reviewer #1: All comments have been addressed

Reviewer #2: All comments have been addressed

2. Is the manuscript technically sound, and do the data support the conclusions?

Reviewer #1: Yes

Reviewer #2: (No Response)

3. Has the statistical analysis been performed appropriately and rigorously?

Reviewer #1: Yes

Reviewer #2: (No Response)

4. Have the authors made all data underlying the findings in their manuscript fully available?

Reviewer #1: Yes

Reviewer #2: (No Response)

5. Is the manuscript presented in an intelligible fashion and written in standard English?

Reviewer #1: Yes

Reviewer #2: (No Response)

Reviewer #1: The authors have satisfactorily addressed all of the comments and concerns raised during the review process. The revised manuscript is substantially improved in terms of clarity, scientific rigor, and translational relevance. I believe the manuscript is now suitable and acceptable for publication.

Reviewer #2: (No Response)

**Do you want your identity to be public for this peer review?** For information about this choice, including consent withdrawal, please see our Privacy Policy

Reviewer #1: **Yes:** Kaushik Banerjee

Reviewer #2: No

---

## [Editor Report · Acceptance letter]

PONE-D-25-53598R1

PLOS One

Dear Dr. Qi,

I'm pleased to inform you that your manuscript has been deemed suitable for publication in PLOS One. Congratulations! Your manuscript is now being handed over to our production team.

Kind regards,

on behalf of

Dr. Tsai-Ching Hsu

Academic Editor

PLOS One